# Mapping Obesity Trends in Saudi Arabia: A Four-Year Description Study

**DOI:** 10.3390/healthcare12202092

**Published:** 2024-10-21

**Authors:** Nora A. Althumiri, Nasser F. Bindhim, Saja A. Al-Rayes, Arwa Alumran

**Affiliations:** 1Informed Decision Making (IDM), Riyadh 13303, Saudi Arabia; nd@idm.sa; 2Sharik Association for Research and Studies, Riyadh 13302, Saudi Arabia; 3Health Information Management and Technology Department, College of Public Health, Imam Abdulrahman bin Faisal University, Dammam 34212, Saudi Arabiaaalumran@iau.edu.sa (A.A.)

**Keywords:** obesity in Saudi Arabia, obesity trend, obesity-related chronic diseases, obesity and comorbidities, obesity prevalence

## Abstract

Objective: Our study aims to map the trends in obesity prevalence over the past four years and to describe the health, behavior, and psychological factors of people living with obesity in Saudi. Method: This is a secondary data analysis using the Sharik Health Indicators Surveillance System (SHISS) from 2020 to 2023. The SHISS dataset comprises cross-sectional telephone interviews carried out quarterly across all administrative regions of Saudi Arabia. Recruitment of participants was restricted to Saudi resident adults only. Results: The study analyzed data from 92,137 participants, with a balanced region and gender distribution. The average age of participants was 36.83 ± 13.68 years. The prevalence of obesity showed minor fluctuations over four years, with the highest at 22.2% in 2020 and lowest at 21.4% in 2023. This study showed that a slight decline in daily smoking rates was observed from 2020 to 2023 across all categories. Participants living with obesity reported a higher consumption of fruits and vegetables compared to their not with obesity counterparts. In addition, participants living with obesity engaged less frequently in physical activities compared to those without obesity. Moreover, people living with obesity have higher incidence rates of depression and anxiety, as well as various of chronic diseases. Conclusions: This study highlights the complex factors affecting obesity prevalence in Saudi Arabia. Despite progress, ongoing health promotion and disease prevention are crucial to address the persistent challenges driven by behavioral and socio-economic factors. Continued surveillance and longitudinal studies are essential to track trends in obesity, smoking, and physical activity, ensuring that health initiatives align with population needs.

## 1. Introduction

Obesity, recognized globally as a chronic disease, is rising in prevalence and has been acknowledged as a worldwide epidemic [1]. This increase is particularly alarming because of the strong epidemiological evidence linking elevated body mass indices (BMIs) to a broad spectrum of chronic conditions [2]. Conditions such as non-alcoholic fatty liver disease (NAFLD), cardiovascular diseases (CVDs), diabetes mellitus, various forms of cancer, musculoskeletal disorders, chronic kidney disease, and mental health disorders are significantly associated with higher BMI levels [3,4,5,6,7,8,9,10,11,12,13,14,15,16,17,18,19,20,21]. These associations are critical because they not only degrade the quality of life of affected individuals but also lead to substantial increases in healthcare costs.

The economic impact of obesity is stark. For instance, individuals with a BMI between 30 and 40 kg/m^2^ encounter healthcare costs that are nearly 50% higher compared to those without obesity-related comorbidities [22,23]. This figure doubles to 100% for individuals with a BMI exceeding 40 kg/m^2^ [23]. Such statistics highlight the severe financial strain on health systems, stemming largely from the treatment of obesity-linked conditions. Additionally, it has been found that a BMI greater than 30 kg/m^2^ can lead to an approximate 37% increase in annual healthcare expenditures [24,25]. This increase is reflective not only of the direct costs associated with medical treatments for obesity and its related diseases but also of the indirect costs such as lost productivity, disability payments, and other health-related inefficiencies. For instance, The World Obesity Atlas 2023, issued by the World Obesity Federation, forecasts that the global economic burden of overweight and obesity could escalate to USD 4.32 trillion annually by 2035, barring significant enhancements in prevention and treatment strategies. Representing nearly 3% of the global GDP, this projected impact is akin to the economic repercussions experienced during the COVID-19 pandemic in 2020 [26].

These financial implications underscore the urgency of addressing obesity not simply as a personal health issue but as a significant public health challenge [27,28,29]. The data suggest a critical need for integrated health strategies that can effectively address and manage obesity on a global scale [29]. Such strategies could include comprehensive public health campaigns aimed at promoting dietary changes, physical activity, and broader lifestyle modifications alongside medical interventions. Moreover, policies that foster environments supportive of healthy choices can also play a crucial role in curbing this epidemic [29,30]. The goal of these initiatives would ultimately be to reduce the incidence of obesity-related diseases and, by extension, the associated healthcare costs that are currently burdening individuals and healthcare systems worldwide.

If current trends continue, by 2035, the majority of the global population—approximately 51%, or over 4 billion people—will be living with either overweight or obesity. Furthermore, about one in four individuals, nearly 2 billion, will be affected by obesity [26]. In the United States of America, data from 2017 to 2020 show that approximately 41.9% of adults were classified as having obesity. Furthermore, the prevalence of obesity is higher in rural communities compared to urban and suburban areas [31]. While obesity levels in Canada have not escalated to those observed in the United States, Canada still exhibits a higher prevalence of obesity compared to other wealthy and developed nations. The rates of obesity have been on a consistent rise in recent years, positioning obesity as one of the major health challenges in the country [32]. In 2022, approximately 30 percent of Canadian adults aged 18 and older were categorized as with obesity, and 35 percent as overweight [32]. Notably, while the percentage of overweight adults remained relatively stable from 2015 to 2022, the prevalence of obesity increased by about four percentage points during the same period [32]. By 2022, there were around 8.7 million adults living with obesity in Canada, in contrast to over ten million adults who were overweight [32]. In the Gulf region, the prevalence of obesity varies significantly among countries. For example, Kuwait has one of the highest rates of obesity globally, surpassing even the United States, with a prevalence of 39.7%. Similarly, the obesity rates in the United Arab Emirates and Qatar are notably high, standing at 27.8% and 46.1%, respectively [33,34]. In contrast, Saudi Arabia shows a relatively lower prevalence of approximately 24%, a figure that aligns with other studies from 2020, which reported obesity rates around 24.7% [35,36].

The lack of a national census in Saudi Arabia specifically addressing the prevalence of obesity in the general population according to age and gender, and how this has changed in recent years, is acknowledged. This gap is primarily due to the limited availability of such comprehensive data. The few existing surveys, such as the National Health Survey by the General Authority for Statistics and the Saudi Health Interview Survey (2019) by the MOH provide valuable insights into obesity trends; however, these are now somewhat outdated, reflecting older data that may not accurately capture recent developments in obesity prevalence [37]. This limitation emphasizes the need for more current and detailed research on obesity in Saudi Arabia and justifies the approach of utilizing available data to explore trends, while also highlighting the potential discrepancies caused by the absence of updated and more frequent national assessments.

Despite the crucial need to monitor obesity trends over time, Saudi Arabia has faced challenges in establishing consistent data collection and measurement methods for tracking obesity rates. Nevertheless, the Sharik Association for Research and Studies, a non-profit entity, has developed a surveillance system to monitor obesity along with other health-related factors. This system offers a consistent framework for data collection, beginning in 2020. Comprehensive details of this model are explained in the methods section of this paper. By leveraging this database, our study aims to map the trends in obesity over the past four years and to describe the health, behavior, and phycological factors of people living with obesity in Saudi Arabia.

## 2. Methods

### 2.1. Data and Sampling

This study conducted a secondary data analysis using data obtained from the Sharik Health Indicators Surveillance System (SHISS) [38]. The SHISS is a biannual, national cross-sectional survey implemented across Saudi Arabia through telephonic interviews [38]. To ensure a representative sample, the SHISS employs a proportional quota sampling technique, which is stratified by age, gender, and region, covering all 13 administrative regions of Saudi Arabia. Additionally, the SHISS incorporates ZDataCloud^®^, an advanced data collection platform designed to reduce sampling bias through automated processes [39]. The dataset analyzed in this study spans the period from 2020 to 2023, including over 95,000 participants [40]. For an in-depth understanding of the methodology implemented by the SHISS, reference may be made to the detailed document published separately by the Sharik Association for Research and Studies website [40].

The sample size was calculated to target a medium effect size of approximately 0.25, ensuring an 80% statistical power and a 95% confidence level [41]. This approach was designed to provide sufficient analytical power for comparisons between regions and to fulfill the sampling quotas. As a result, each quota necessitated at least 134 participants, with each region requiring 536 participants. This sampling strategy resulted in a total of 6968 participants for each wave from 2020 to 2023.

### 2.2. Participant Recruitment

The recruitment of participants was restricted to Arabic-speaking residents of Saudi Arabia who were 18 years of age or older. The Sharik Association for Health Research generated a list of random mobile phone numbers to locate potential participants [42]. By 2023, The Sharik database included over 260,000 participants from Saudi Arabia’s 13 regions who had expressed interest in future research, continuing to expand [42]. Upon consent, the interviewer verified the participant’s eligibility before commencing the interview.

### 2.3. Measures

#### 2.3.1. Demographics

Demographic information collected from the Sharik Health Indicators Surveillance System (SHISS) database included age, gender, income, and education level. Obesity was defined based on a Body Mass Index (BMI) of 30 kg/m^2^ or above, which was calculated using self-reported height and weight data [43].

#### 2.3.2. Questionnaire Design and Data Model

Upon verbal consent, participants disclosed their age and region for eligibility assessment. Subsequently, the data collectors enter the participant’s age, gender, and region along with any major chronic diseases present directly in the electronic system, ZDataCloud [39]. Additionally, their primary behavioral and intermediate risk factors, as recommended by the World Health Organization (WHO) and the Centers for Disease Control and Prevention (CDC), were evaluated [43,44,45]. As depicted in the SHISS data model (Figure 1), the dataset encompasses behavioral risk factors such as diet, physical activity, and tobacco use (including cigarettes, waterpipes, and e-cigarettes). It also includes diagnosed and treated intermediate risk factors like hypertension and hypercholesterolemia, obesity quantified by Body Mass Index (BMI) derived from self-reported height and weight measurements, and major chronic diseases currently under treatment, such as diabetes, heart disease, stroke, cancer, and chronic respiratory disease. In addition, the dataset recorded the presence of diagnosed genetic diseases, classified as a nonmodifiable risk factor [43,44,45].

### 2.4. Data Analysis

In this study, descriptive statistics were expressed using frequencies and proportions. The data, collected via ZDataCloud^®^, exhibited no instances of missing information [39]. Statistical analyses were carried out using SPSS version 29, and findings were documented following the Strengthening the Reporting of Observational Studies in Epidemiology (STROBE) guidelines for cross-sectional studies [46].

### 2.5. Ethical Considerations

This project received approval from the Sharik Institutional Review Board (approval no. 2021-2), consistent with the national research ethics law and regulations of Saudi Arabia. Consent from participants was secured verbally during the interview process.

## 3. Results

A total of 92,137 participants across all 13 administrative regions completed the interview. Of these, 50.1% were female, with an average age of 36.83 ± 13.68 years (range: 18–90 years), and the median age was 36. The prevalence of obesity is slightly decreasing, recorded at 22.2% in 2020, 22.1% in 2021, 21.2% in 2022, and 21.4% in 2023. Higher prevalences were found in female across the four years. Table 1 displays the demographic distribution of the sample based on obesity status.

Table 2 illustrates the distribution of participants across both groups (with and without obesity) for eight chronic diseases. Participants with obesity exhibited higher incidence rates for all types of diseases, including hypertension, hypercholesterolemia, type 2 diabetes, cardiovascular diseases, lung diseases, cancer, stroke, and genetic disorders. Over the four-year period, the percentage of individuals diagnosed with hypertension and obesity decreased from 25.4% in 2020 to 20.2% in 2023, with a corresponding reduction in individuals without obesity from 12.8% to 8.8%. A similar trend was observed for hypercholesterolemia, with obesity-related diagnoses decreasing from 25.0% in 2020 to 21.6% in 2023. In contrast, diagnoses of hypercholesterolemia in individuals without obesity remained relatively stable, ranging from 9.5% to 10.7%. In the case of type 2 diabetes, the proportion of individuals with obesity diagnosed with the condition decreased from 22.5% in 2020 to 18.8% in 2023, while the percentage among those without obesity remained stable around 10%. Cardiovascular diseases followed a similar trend, with the proportion of individuals with diagnosed obesity decreasing from 8.4% in 2020 to 7.3% in 2023. In individuals without obesity, the prevalence of cardiovascular diseases remained steady at around 4%. Respiratory diseases showed a slight decline among individuals with obesity, decreasing from 11.4% in 2020 to 9.4% in 2023, while the proportion of individuals without obesity remained around 7%. Cancer diagnoses among individuals with obesity also showed a small decrease from 2.6% in 2020 to 2.0% in 2023. Stroke diagnoses remained relatively low but stable, with the percentage of individuals with obesity diagnosed with stroke slightly decreasing from 3.0% in 2020 to 2.1% in 2023.

Lastly, genetic diseases were consistently more prevalent among individuals with obesity compared to those without, though the percentage decreased from 9.6% in 2020 to 6.6% in 2023. Individuals without obesity showed a steady prevalence of around 4–6% during the same period.

Table 3 presents the distribution of behavioral characteristics by obesity status, detailing patterns in smoking (cigarettes, waterpipes, and vaping), engagement in physical activities (both intensive and moderate), consumption of healthy foods (vegetables and fruits), and mental health risks (depression and anxiety). Smoking habits showed that the majority of participants, both with and without obesity, reported never smoking cigarettes, with a slight increase in the proportion of non-smokers among those with obesity, rising from 84.0% in 2020 to 86.7% in 2023. Similarly, waterpipe smoking (shisha) and vaping remained relatively low, with the majority of participants reporting never using these substances. Among those with obesity, 85.5% had never smoked shisha in 2020, and this percentage increased slightly to 87.7% in 2023. Vaping behavior followed a similar trend, with 92.7% of participants with obesity in 2020 reporting never vaping, compared to 89.7% in 2023.

Weight measurement habits indicate that a significant proportion of participants, both with and without obesity, reported having weighed themselves recently. In 2021, 25.4% of participants without obesity and 22.9% with obesity had measured their weight in the past week, with these percentages remaining stable in subsequent years. However, more than 30% of participants with obesity had not weighed themselves in over a month by 2023.

Dietary habits, as measured by acceptable levels of vegetable and fruit consumption, showed slight variations over the years. Among participants with obesity, 84.0% reported not consuming acceptable levels of vegetables in 2020, compared to 85.5% in 2023. Fruit consumption followed a similar pattern, with 88.8% of participants with obesity reporting inadequate fruit intake in 2020, which slightly improved to 92.2% in 2023. Combining both measures of vegetable and fruit consumption, a large proportion of participants, regardless of obesity status, consistently reported not meeting recommended levels throughout the study period.

Physical activity levels showed that a high percentage of participants with obesity were below the recommended levels of intensive and moderate physical activity. In 2020, 85.5% of participants with obesity were below the recommended level of intensive physical activity, increasing to 91.4% by 2023. A similar trend was observed in moderate physical activity, with 83.2% of participants with obesity falling below recommended levels in 2020, and 87.8% in 2023. When combining both measures of physical activity, 67.5% of participants with obesity in 2020 were below recommendations, rising to 84.0% in 2023.

Mental health indicators, including risk of depression (measured by PHQ-2) and anxiety (measured by GAD-2), were added to the survey in 2021 and 2022, respectively. In 2021, 20.3% of participants with obesity were at high risk of depression, a figure that remained consistent at 20.7% in 2023. The proportion of participants with obesity at high risk of anxiety was 21.7% in 2022, slightly decreasing to 20.1% in 2023.

Figure 2 illustrates the visualization of trends related to obesity, modifiable risk factors, intermediate risk factors, and major chronic diseases from 2020 to 2023.

## 4. Discussion

This study investigated obesity trends between 2020 and 2023, and described the health, behaviors, and psychological factors of individuals living with obesity. The results revealed nearly stable levels of obesity prevalence between 2020 and 2023 in Saudi Arabia across both genders. The 20–29 age group showed a consistent decrease in obesity during this period, with a total reduction of 3.7%. Smoking cigarettes and waterpipe use have decreased, while e-cigarette use is increasing among people living with obesity. In terms of behaviors, there is an overall low consumption of fruits and vegetables and low physical activity, both of which are highly prevalent among people living with obesity, with slight variations between the years. However, the prevalence of chronic diseases, including hypertension, hypercholesterolemia, cardiovascular disease, diabetes, stroke, and respiratory diseases, is significantly higher among people living with obesity compared to their non-obese counterparts across the study years.

Detecting a statistically significant change in obesity prevalence may require several years or even decades, depending on the intensity and reach of interventions. For instance, comprehensive community programs and national policies might begin to show effects on obesity prevalence within 5 to 10 years. In some cases, noticeable changes might take longer to manifest in population-level data. However, Saudi Arabia recently announced an increase in life expectancy to 77.6 years [47]. This increase may result from multiple public initiatives and policies implemented by Saudi health sectors aimed at reducing salt and hydrogenated oil consumption, mandating the declaration of total calories in all products as well as prepared and served meals, expanding physical activity rehabilitation centers, and enhancing infrastructure and awareness related to walking. If these initiatives directly affect the prevalence of obesity and continue to contribute to its decline, this could have a substantial impact on the health and economic systems of Saudi Arabia.

However, the prevalence of hypertension, hypercholesterolemia, type 2 diabetes, cardiovascular diseases, lung diseases, stroke, and genetic disorders was notably higher, and in some cases doubled, among participants living with obesity over the study period. These results align with international studies and reinforce the link between obesity and these health conditions, transcending sociodemographic differences across countries [3,6,11,14,24,48,49]. Furthermore, these findings emphasize the concurrent prevalence of obesity and chronic health issues, which could potentially exacerbate the health burden on individuals and diminish outcomes in disease management. Given these implications, prioritizing these populations is crucial for clinical practices and economic impact in Saudi Arabia.

This study identified a modest decrease in smoking rates among all participants. Interestingly, individuals without obesity were more likely to smoke compared to those with obesity, a finding that is consistent with existing literature [50]. Additionally, data from the Centers for Disease Control and Prevention (CDC) highlight a reduction in smoking rates among American adults, from 20.9% in 2005 to 11.5% in 2021 [50]. Similarly, the National Drug Strategy Household Survey (NDSHS) documented a significant decline in daily smoking among individuals aged 14 and over, from 24% in 1991 to 8.3% in the period 2022–2023 [51]. These trends underscore global efforts to reduce smoking, as evidenced by countries’ adherence to the WHO Framework Convention on Tobacco Control [52]. In the beginning of 2020, the health sector in Saudi Arabia launched significant campaigns aimed at reducing smoking and tobacco use, including the implementation of plain packaging. These measures may have contributed to the observed reduction in smoking rates during this period.

In the context of weight management, our study observed that individuals living with obesity were less inclined to engage in self-weighing practices compared to their non-obese counterparts. This trend is notable, as self-weighing has been implicated as a protective factor to obesity in several preliminary studies [53,54]. Engaging in frequent self-weighing has been consistently associated with several positive outcomes, including moderate weight loss, minimized risk of weight regain, and prevention of initial weight gain in adult populations. The literature suggests that self-weighing may serve as a reflective tool that encourages individuals to remain conscious of their weight changes, thereby fostering adherence to weight management practices [53]. However, despite these findings, the mechanisms through which self-weighing influences weight outcomes are not fully understood and warrant further investigation. Expanding the scope of research in this area could offer deeper insights into the behavioral, psychological, and physiological aspects of self-weighing and its effectiveness as a preventive strategy against obesity. This could potentially guide the development of tailored interventions that enhance the efficacy of self-weighing as part of comprehensive obesity management programs.

Regarding physical activity, our study found that individuals without obesity engaged in slightly higher levels of physical exercise compared to those with obesity. This disparity in physical activity is critical to understanding obesity management, as sedentary lifestyles—characterized by insufficient physical activity, limited body movement, and reduced muscle contractions—are widely recognized as contributing factors to the development and persistence of obesity [55]. The correlation between reduced physical activity and increased obesity prevalence has been consistently observed across numerous studies [55,56]. These studies highlight that regular physical exercise not only helps in reducing body weight but also mitigates the risks associated with chronic diseases linked to obesity [49,57]. Saudi Arabia first permitted gyms for women in 2012, and over the subsequent decade, the number of gyms accessible to both genders has expanded to 2295. This increase and diversification in gym memberships may play a role in enhancing engagement in physical activities. Nevertheless, additional research is needed to investigate the reasons why individuals with obesity are less likely to participate in physical activity.

Furthermore, mental health, assessed by the risks of depression and anxiety, was found to be slightly elevated among individuals living with obesity. Extensive research indicates that people living with obesity have an 18% to 55% increased likelihood of developing depression. Conversely, the probability of being classified as having obesity is raised by 37% to 58% in individuals with depression [58]. Psychiatric evaluations may thus be a critical component of comprehensive obesity care, warranting further investigation to potentially enhance both therapeutic and functional outcomes. The emerging understanding of the bidirectional relationship between psychiatric disorders and obesity suggests that treating psychiatric conditions may concurrently reduce the burden of obesity, and vice versa. Future research should focus on defining the relationship between the severity of obesity and specific clinical subtypes of depression and anxiety, considering the diverse manifestations of these conditions. Additionally, researchers should prioritize methodological considerations, such as utilizing objective measurements of obesity rather than self-reported data and accounting for potential confounders like physical comorbidities, to elucidate the intricate connections between obesity and psychiatric disorders.

Several initiatives have been implemented in recent years to address this growing public health concern. One of the key drivers is the Saudi Vision 2030 initiative, which includes the Quality-of-Life Program that promotes physical activity by increasing access to recreational spaces and fitness facilities to encourage healthier lifestyles [59]. The National Strategy for Diet and Physical Activity (2018–2022), launched by the Ministry of Health (MoH), also plays a vital role by focusing on educational campaigns that promote healthy eating habits and physical activity in schools, workplaces, and communities [60]. Another significant policy is the introduction of a sugar tax in 2017, which imposed a 50% tax on sugar-sweetened beverages and a 100% tax on energy drinks, aimed at reducing the consumption of sugary products, a major contributor to obesity [61].

In addition, the Saudi Food and Drug Authority (SFDA) has established clear labeling regulations, requiring food products to display calorie, fat, and sugar content, helping consumers make informed dietary choices. These regulations also restrict the marketing of unhealthy food, especially to children [62]. The Ministry of Health has launched various health promotion and education programs, including public awareness campaigns and school-based initiatives, to encourage healthier behaviors and raise awareness about the risks of obesity [63,64]. Furthermore, the government has expanded public health infrastructure to manage obesity-related conditions, such as setting up specialized clinics for obesity management and supporting bariatric surgery for eligible patients [64]. While these policies represent a comprehensive approach to combating obesity in Saudi Arabia, their long-term effectiveness remains to be fully assessed as the country continues to face the challenges of rising obesity rates.

This investigation possesses both strengths and weaknesses. The key strengths lie in its population-based, nationwide approach, featuring a substantial sample size and high-quality data. Nonetheless, the cross-sectional design of the study precludes the determination of causality. Consequently, it is not feasible to ascertain trends, leaving the question of whether obesity rates are genuinely declining unresolved. A principal limitation is the reliance on self-reported data, which may introduce bias into the findings.

## 5. Conclusions

This study has provided insights into the dynamics of obesity prevalence and has described the health, behavioral, and physiological factors within Saudi Arabia over recent years. These factors may collectively contribute to tackling obesity, though understanding the prevalence of obesity remains a complex challenge influenced by multiple behavioral and socio-economic factors. Continued commitment to health promotion and disease prevention strategies is essential for sustaining the progress made thus far and for furthering the impact on the health and economic systems of Saudi Arabia. To minimize the clinical and nutritional impact of obesity on Saudi Arabia, targeted interventions should include enhanced public health campaigns focusing on nutrition education, particularly in schools and community settings, to instill healthy eating habits from an early age. Expanding access to diet counseling and obesity management services, such as weight-loss clinics and nutritionists, will provide individuals with the tools needed to make healthier choices. Furthermore, increasing public access to physical activity facilities and incorporating exercise programs into the routine healthcare system can further aid in obesity prevention. On a broader scale, implementing stronger regulations on unhealthy food marketing, especially to children, and encouraging healthier food options in public institutions are critical steps.

Enhanced surveillance and more longitudinal studies are needed to track the long-term trends in obesity, smoking, and physical activity, ensuring that health initiatives are effectively aligned with the needs of the population. Through a multi-pronged approach that combines education, clinical support, and regulatory measures, the clinical and nutritional impact of obesity on the country’s healthcare system can be reduced, promoting better long-term health outcomes.

## Figures and Tables

**Figure 1 healthcare-12-02092-f001:**
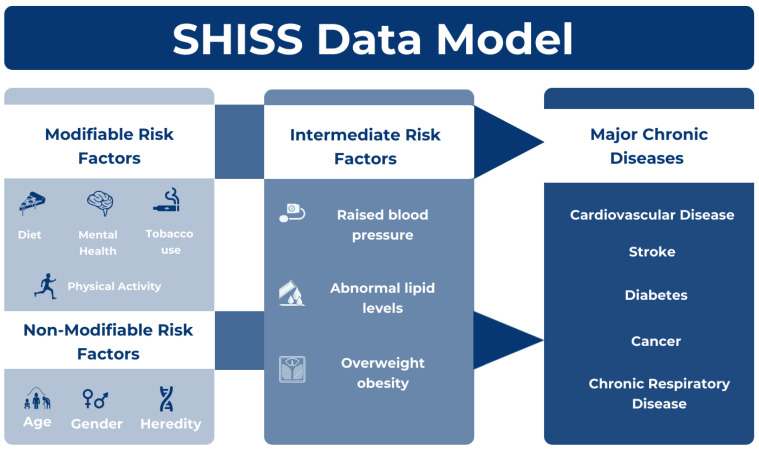
The Sharik Health Indicators Surveillance System (SHISS) Model [38].

**Figure 2 healthcare-12-02092-f002:**
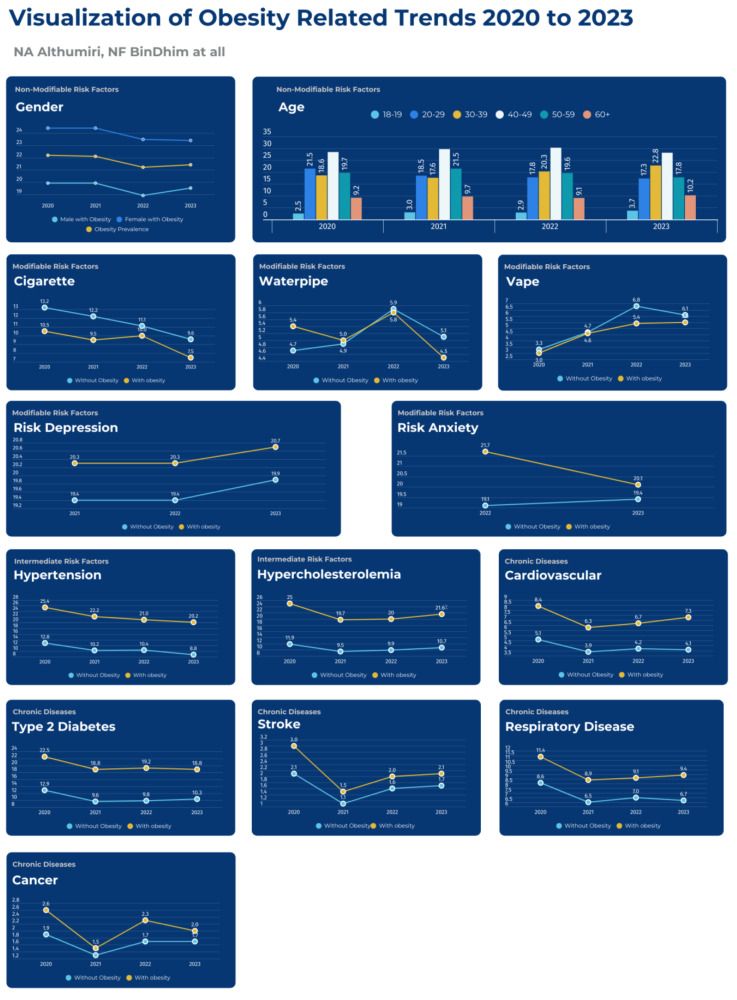
Visualization of trends related to obesity from 2020 to 2023.

**Table 1 healthcare-12-02092-t001:** Demographic characteristics of participants distributed by obesity status.

Variable	2020		2021		2022		2023	
	Without Obesity n (%)	With Obesity n (%)	Without Obesity n (%)	With Obesity n (%)	Without Obesity n (%)	With Obesity n (%)	Without Obesity n (%)	With Obesity n (%)
**Gender**								
Male	9155 (80.1)	2278 (19.9)	11202 (80.1)	2788 (19.9)	1105 (81.1)	2568 (18.9)	5600 (80.5)	1358 (19.5)
Female	8813 (75.6)	2852 (24.4)	10539 (75.6)	3393 (24.4)	10394 (76.5)	3187 (23.5)	5330 (76.6)	1625 (23.4)
Total	17968 (77.8)	5130 (22.2)	21741 (77.9)	6181 (22.1)	21449 (78.8)	5755 (21.2)	10930 (78.6)	2983 (21.4)
**Age Group**								
18–19	929 (5.2)	127 (2.5)	1228 (5.6)	185 (3.0)	1174 (5.5)	167 (2.9)	678 (6.2)	111 (3.7)
20–29	6733 (37.5)	1104 (21.5)	8144 (37.5)	1146 (18.5)	7711 (36.0)	1025 (17.8)	3726 (34.1)	516 (17.3)
30–39	3806 (21.2)	955 (18.6)	4557 (21.0)	1088 (17.6)	4910 (22.9)	1171 (20.3)	2574 (23.5)	679 (22.8)
40–49	3432 (19.1)	1461 (28.5)	4324 (19.9)	1834 (29.7)	4294 (20.0)	1741 (30.3)	2187 (20.0)	841 (28.2)
50–59	1934 (10.8)	1009 (19.7)	2236 (10.3)	1331 (21.5)	2174 (10.1)	1126 (19.6)	1079 (9.9)	531 (17.8)
60 and above	1134 (6.3)	474 (9.2)	1252 (5.8)	597 (9.7)	1186 (5.5)	525 (9.1)	686 (6.3)	305 (10.2)
**Administrative Region**								
Al-Jouf	1302 (71.7)	488 (27.3)	1615 (74.7)	546 (22.2)	1668 (77.5)	484 (22.5)	816 (75.8)	261 (24.2)
Northern Boarders	1397 (80.1)	348 (19.9)	1720 (79.4)	445 (20.6)	1592 (73.9)	562 (26.1)	841 (77.8)	240 (22.2)
Tabuk	1386 (79.8)	351 (20.2)	1740 (80.3)	427 (19.7)	1800 (83.1)	365 (16.9)	866 (80.2)	214 (19.8)
Hail	1410 (79.0)	374 (21.0)	1658 (76.6)	507 (23.4)	1612 (74.5)	551 (25.5)	875 (81.2)	203 (18.8)
Almadinah	1470 (76.8)	445 (23.2)	1705 (79.9)	430 (20.1)	1753 (81.2)	406 (18.8)	873 (80.8)	208 (19.2)
Qassim	1364 (77.5)	397 (22.5)	1691 (78.1)	475 (21.9)	1279 (78.9)	342 (21.1)	831 (76.9)	250 (23.1)
Macca	1363 (75.7)	438 (24.3)	1617 (74.8)	546 (25.2)	1719 (79.3)	448 (20.7)	799 (73.4)	289 (23.1)
Riyadh	1352 (73.5)	488 (26.5)	1602 (73.7)	572 (26.3)	1611 (74.4)	554 (25.6)	799 (73.7)	285 (26.3)
Eastern Region	1332 (73.3)	484 (26.7)	1636 (75.6)	527 (24.4)	1636 (75.6)	528 (24.4)	817 (75.6)	264 (24.4)
Albaha	1537 (86.1)	249 (13.9)	1864 (86.1)	301 (13.9)	1898 (87.7)	265 (12.3)	922 (85.4)	158 (14.6)
Asir	1512 (77.4)	441 (22.6)	1624 (75.1)	538 (24.9)	1645 (76.1)	517 (23.9)	823 (75.8)	263 (24.2)
Jazan	1478 (80.3)	362 (19.7)	1747 (80.6)	420 (19.4)	1806 (83.6)	353 (16.4)	898 (82.9)	185 (17.1)
Najran	1065 (80.1)	265 (19.9)	1552 (77.3)	447 (22.7)	1430 (79.0)	380 (21.0)	770 (82.5)	163 (17.5)
Total	17968 (77.8)	5130 (22.2)	21741 (77.9)	6181 (22.1)	21449 (78.8)	5755 (21.2)	10930 (78.6)	2983 (21.4)
**Income Level ***								
Less than 5000 SR	-	-	7937 (36.5)	1965 (31.8)	6498 (30.3)	1449 (25.2)	3816 (34.9)	801 (26.9)
Between 5000 SR and 8000 SR	-	-	3094 (14.2)	852 (13.8)	3252 (15.2)	878 (15.3)	1673 (15.3)	469 (15.7)
Between 8001 SR and 11,000 SR	-	-	2379 (10.9)	686 (11.1)	2481 (11.6)	658 (11.4)	1213 (11.1)	344 (11.5)
Between 11,001 SR and 13,000 SR	-	-	1629 (7.5)	529 (8.6)	1535 (7.2)	484 (8.4)	683 (6.2)	238 (8.0)
Between 13,001 SR and 16,000 SR	-	-	1312 (6.0)	503 (8.1)	1432 (6.7)	456 (7.9)	708 (6.5)	246 (8.2)
Between 16,001 SR and 20,000 SR	-	-	1026 (4.7)	406 (6.6)	981 (4.6)	341 (5.9)	501 (4.6)	195 (6.5)
More than 20,000 SR	-	-	697 (3.2)	217 (3.8)	784 (3.6)	268 (4.3)	357 (3.3)	107 (3.6)
No stable income	-	-	4573 (21.3)	1272 (22.1)	3580 (16.5)	972 (15.7)	1979 (18.1)	583 (19.5)
**Educational Level ***								
Less Than Bachelor	-	-	10747 (49.4)	3554 (57.5)	10663 (49.7)	3229 (56.1)	5534 (50.6)	1669 (56.0)
Bachelor and above	-	-	10994 (50.6)	2627 (42.5)	10786 (50.3)	2526 (43.9)	5396 (49.4)	1314 (44.0)

* Income and Education variables were not included in the data collection in 2020.

**Table 2 healthcare-12-02092-t002:** Chronic diseases incidence stratified by obesity status.

Variable	2020		2021		2022		2023	
	Without Obesity n (%)	With Obesity n (%)	Without Obesity n (%)	With Obesity n (%)	Without Obesity n (%)	With Obesity n (%)	Without Obesity n (%)	With Obesity n (%)
Diagnosed with Hypertension	2304 (12.8)	1251 (25.4)	2220 (10.2)	1372 (22.2)	2226 (10.4)	1206 (21.0)	966 (8.8)	604 (20.2)
Diagnosed with Hypercholesterolemia	2144 (11.9)	1282 (25.0)	2063 (9.5)	1219 (19.7)	2122 (9.9)	1149 (20.0)	1173 (10.7)	643 (21.6)
Diagnosed with Type 2 Diabetes	2130 (12.9)	1152 (22.5)	2091 (9.6)	1162 (18.8)	2110 (9.8)	1106 (19.2)	1127 (10.3)	561 (18.8)
Cardiovascular Diseases	916 (5.1)	429 (8.4)	846 (3.9)	392 (6.3)	901 (4.2)	399 (6.7)	452 (4.1)	219 (7.3)
Respiratory disease	1514 (8.6)	586 (11.4)	1419 (6.5)	548 (8.9)	1494 (7.0)	526 (9.1)	730 (6.7)	279 (9.4)
Cancer	339 (1.9)	132 (2.6)	280 (1.3)	92 (1.5)	360 (1.7)	131 (2.3)	191 (1.7)	61 (2.0)
Stroke	370 (2.1)	156 (3.0)	249 (1.1)	93 (1.5)	346 (1.6)	117 (2.0)	184 (1.7)	62 (2.1)
Genetic Disease	1163 (6.5)	495 (9.6)	955 (4.4)	431 (7.0)	991 (4.6)	362 (6.3)	512 (4.7)	198 (6.6)

**Table 3 healthcare-12-02092-t003:** Behavioral characteristics of participants distributed by obesity status.

Variable	2020		2021		2022		2023	
	Without Obesity n (%)	With Obesity n (%)	Without Obesity n (%)	With Obesity n (%)	Without Obesity n (%)	With Obesity n (%)	Without Obesity n (%)	With Obesity n (%)
Smoking Cigarettes								
Never	14,232 (79.2)	4309 (84.0)	17,611 (81.0)	5286 (85.5)	17,530 (81.7)	4897 (85.1)	9194 (84.1)	2587 (86.7)
Yes, occasionally	1367 (7.6)	284 (5.5)	1473 (6.8)	307 (5.0)	1543 (7.2)	280 (4.9)	691 (6.3)	171 (5.7)
Yes, daily	2369 (13.2)	537 (10.5)	2657 (12.2)	588 (9.5)	2376 (11.1)	578 (10.0)	1045 (9.6)	225 (7.5)
Smoking Waterpipe (Shishah)								
Never	15,268 (85.0)	4403 (85.5)	18,618 (85.6)	5413 (87.6)	18,240 (85.0)	4963 (86.2)	9513 (87.0)	2615 (87.7)
Yes, occasionally	1848 (10.3)	452 (8.8)	2061 (9.5)	456 (7.4)	1951 (9.1)	457 (7.9)	863 (7.9)	234 (7.8)
Yes, daily	852 (4.7)	275 (5.4)	1062 (4.9)	312 (5.0)	1258 (5.9)	335 (5.8)	554 (5.1)	134 (4.5)
Smoking Vape								
Never	16,420 (91.4)	4758 (92.7)	19,427 (89.4)	5646 (91.3)	18,520 (86.3)	5118 (88.9)	9585 (87.7)	2677 (89.7)
Yes, occasionally	957 (5.3)	218 (4.2)	1296 (6.0)	251 (4.1)	1470 (6.9)	329 (5.7)	675 (6.2)	142 (4.8)
Yes, daily	591 (3.3)	154 (3.0)	1018 (4.7)	284 (4.6)	1459 (6.8)	308 (5.4)	670 (6.1)	164 (5.5)
Last Time for Weight Measurement *								
This Week	-	-	5523 (25.4)	1417 (22.9)	5398 (25.2)	1397 (24.3)	2756 (25.2)	755 (25.3)
Two Weeks Ago	-	-	4697 (21.6)	1263 (20.4)	4866 (22.7)	1269 (22.1)	2418 (22.1)	644 (21.6)
Last Month	-	-	4975 (22.9)	1424 (23.0)	4799 (22.4)	1275 (22.2)	2488 (22.8)	616 (20.7)
More Than a Month Ago	-	-	6546 (30.1)	2077 (33.6)	6386 (29.8)	1814 (31.5)	3268 (29.9)	968 (32.5)
Acceptable Level of Vegetables (ALV)								
No	15,463 (86.1)	4311 (84.0)	19,098 (87.8)	5354 (86.6)	18,323 (85.4)	4827 (83.9)	9492 (86.8)	2551 (85.5)
Yes	2505 (13.9)	819 (16.0)	2643 (12.2)	827 (13.4)	3126 (14.6)	928 (16.1)	1438 (13.2)	432 (14.5)
Acceptable Level of Fruits (ALF)								
No	16,287 (90.6)	4557 (88.8)	19,523 (89.8)	5517 (89.3)	19,515 (91.0)	5169 (89.8)	10,112 (92.5)	2749 (92.2)
Yes	1618 (9.4)	573 (11.2)	2218 (10.2)	664 (10.7)	1934 (9.0)	586 (10.2)	818 (7.5)	234 (7.8)
Combining (ALV) and (ALF)								
No	17,032 (94.8)	4835 (49.2)	20,574 (94.6)	5844 (94.5)	20,120 (93.8)	5357 (93.1)	10,328 (94.5)	2983 (94.9)
Yes	936 (5.2)	295 (5.8)	1167 (5.4)	337 (5.5)	1329 (6.2)	398 (6.9)	602 (5.5)	153 (5.1)
Intensive Physical Activity								
Below the Recommendation	14,263 (79.4)	4388 (85.5)	18,277 (84.1)	5518 (89.3)	18,320 (85.4)	5220 (90.7)	9359 (85.6)	2725 (91.4)
As Recommended	3705 (20.6)	742 (14.5)	3464 (15.9)	663 (10.7)	3129 (14.6)	353 (9.3)	1517 (14.4)	258 (8.6)
Moderate Physical Activity								
Below the Recommendation	13,954 (77.7)	4266 (83.2)	17,254 (79.4)	5216 (84.4)	17,391 (81.1)	4910 (85.3)	8971 (82.1)	2619 (87.8)
As Recommended	4487 (20.6)	965 (15.6)	4487 (20.6)	965 (15.6)	4058 (18.9)	845 (14.7)	1959 (17.9)	364 (12.2)
Combined Physical Activity								
Below the Recommendation	12,329 (68.6)	3924 (67.5)	15,856 (72.9)	4922 (79.6)	16,036 (74.8)	4671 (81.2)	8355 (76.4)	2506 (84.0)
As Recommended	5639 (31.4)	1206 (23.5)	5885 (27.1)	1259 (20.4)	5413 (25.2)	1084 (18.8)	2575 (23.6)	477 (16.0)
At High Risk of Depression (PHQ2) *								
No	-	-	17,529 (80.6)	4925 (79.7)	17,282 (80.6)	4589 (79.7)	8756 (80.1)	2365 (79.3)
Yes	-	-	4212 (19.4)	1256 (20.3)	4167 (19.4)	1166 (20.3)	2174 (19.9)	618 (20.7)
At High Risk of Anxiety (GAD2) **								
No	-	-	-	-	12,997 (80.9)	3336 (78.3)	8809 (80.6)	2384 (79.9)
Yes	-	-	-	-	3061 (19.1)	923 (21.7)	2121 (19.4)	599 (20.1)

* Variable was not included in the data collection during 2020. ** Anxiety risk variable was included in the SHISS in 2022.

## Data Availability

Data is contained within the article.

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
