# Peer review of "Mapping Obesity Trends in Saudi Arabia: A Four-Year Description Study"

_healthcare, 2024, doi:10.3390/healthcare12202092_

Round 1
Reviewer 1 Report
Comments and Suggestions for Authors
This study aims to map the trends of obesity prevalence over the past four years.
My comments:
1. In the introduction, please provide more information about a national census already carried out in Saudi Arabia addressing the prevalence of obesity in the general population according to age or gender and how this has changed in recent years. Since the aim of the study is to describe obesity status, this information is necessary. Or the lack of them justified this study.
2. The anthropometric measures to calculate BMI were collected or reported. Please better clarify in the methods section.
3. Considering the sample size and the data collected, I suggest correlation and association analyses between the variables collected and not just descriptive analyses.
4. In the discussion, please include data on public health policies that aim to mitigate the effects of the increase in obesity in the country.
5. In the conclusion section, what the authors suggest as a way to minimize the clinical and nutritional impact of obesity on the country.
Author Response
- In the introduction, please provide more information about a national census already carried out in Saudi Arabia addressing the prevalence of obesity in the general population according to age or gender and how this has changed in recent years. Since the aim of the study is to describe obesity status, this information is necessary. Or the lack of them justified this study.
Author Response: Thank you for this valuable comment. Unfortunately, Saudi Arabia does not have a traditional national census specifically focused on obesity or other health conditions. However, several national surveys and studies have provided robust data on obesity prevalence in the country. However, the manuscript has been updated as the following
" The lack of a national census in Saudi Arabia specifically addressing the prevalence of obesity in the general population according to age and gender, and how this has changed in recent years, is acknowledged. This gap is primarily due to the limited availability of such comprehensive data. The few existing surveys, such as the National Health Survey by General Authority for Statistics and the Saudi Health Interview Survey (2019) by MOH provide valuable insights into obesity trends; however, these are now somewhat outdated, reflecting older data that may not accurately capture recent developments in obesity prevalence [37]. This limitation emphasizes the need for more current and detailed research on obesity in Saudi Arabia and justifies the approach of utilizing available data to explore trends, while also highlighting the potential discrepancies caused by the absence of updated and more frequent national assessments."
- The anthropometric measures to calculate BMI were collected or reported. Please better clarify in the methods section.
Author Response: Noted with thanks. The BMI were self-reported. This were added in the method section. The manuscript has been updated as the following
"Obesity quantified by body mass index (BMI) derived from self-reported height and weight measurements"
- Considering the sample size and the data collected, I suggest correlation and association analyses between the variables collected and not just descriptive analyses.
Author Response: Thank you for your valuable comments. This paper aimed to explore the trend of obesity over a 4-year period. Previously, we published a paper on obesity and its associated factors in Saudi Arabia. In that study, we conducted association analyses and found similar results with this study. Therefore, we decided not to include the association analyses in this paper to avoid duplication, as it would not add new insights to the existing literature. The focus of this paper is on describing the trends over time rather than replicating previously reported associations. The paper can be found (https://www.mdpi.com/2227-9032/9/3/311)
- In the discussion, please include data on public health policies that aim to mitigate the effects of the increase in obesity in the country.
Author Response: Thank you for the insightful comment. We have now included information on the various public health policies and initiatives introduced in Saudi Arabia to address the rising prevalence of obesity. The manuscript has been updated as the following
" Several initiatives have been implemented in recent years to address this growing public health concern. One of the key drivers is the Saudi Vision 2030 initiative, which includes the Quality-of-Life Program that promotes physical activity by increasing access to recreational spaces and fitness facilities to encourage healthier lifestyles [57]. The National Strategy for Diet and Physical Activity (2018-2022), launched by the Ministry of Health (MoH), also plays a vital role by focusing on educational campaigns that promote healthy eating habits and physical activity in schools, workplaces, and communities[58]. Another significant policy is the introduction of a sugar tax in 2017, which imposed a 50% tax on sugar-sweetened beverages and a 100% tax on energy drinks, aimed at reducing the consumption of sugary products, a major contributor to obesity[59].
In addition, the Saudi Food and Drug Authority (SFDA) has established clear labeling regulations, requiring food products to display calorie, fat, and sugar content, helping consumers make informed dietary choices. These regulations also restrict the marketing of unhealthy food, especially to children[60]. The Ministry of Health has launched various health promotion and education programs, including public awareness campaigns and school-based initiatives, to encourage healthier behaviors and raise awareness about the risks of obesity[61, 62]. Furthermore, the government has expanded public health infrastructure to manage obesity-related conditions, such as setting up specialized clinics for obesity management and supporting bariatric surgery for eligible patients[62]. While these policies represent a comprehensive approach to combating obesity in Saudi Arabia, their long-term effectiveness remains to be fully assessed as the country continues to face the challenges of rising obesity rates."
- In the conclusion section, what the authors suggest as a way to minimize the clinical and nutritional impact of obesity on the country.
Author Response: Thank you for the insightful comment. we have updated the manuscript as the following:
"This study has provided insights into the dynamics of obesity prevalence and described the health, behavioral, and physiological factors within Saudi Arabia over recent years. These factors collectively may contribute to tackling obesity, though the prevalence of obesity remains a complex challenge influenced by multiple behavioral and socio-economic factors. Continued commitment to health promotion and disease prevention strategies is essential for sustaining the progress made thus far and for furthering the impact on the health and economic systems of Saudi Arabia. To minimize the clinical and nutritional impact of obesity on the Saudi Arabia, targeted interventions should include enhanced public health campaigns focusing on nutrition education, particularly in schools and community settings, to instill healthy eating habits from an early age. Expanding access to diet counseling and obesity management services, such as weight-loss clinics and nutritionists, will provide individuals with the tools needed to make healthier choices. Furthermore, increasing public access to physical activity facilities and incorporating exercise programs into the routine healthcare system can further aid in obesity prevention. On a broader scale, implementing stronger regulations on unhealthy food marketing, especially to children, and encouraging healthier food options in public institutions are critical steps.
Enhanced surveillance and more longitudinal studies are needed to track the long-term trends in obesity, smoking, and physical activity, ensuring that the health initiatives are effectively aligned with the needs of the population. Through a multi-pronged approach that combines education, clinical support, and regulatory measures, the clinical and nutritional impact of obesity on the country's healthcare system can be reduced, promoting better long-term health outcomes."
Reviewer 2 Report
Comments and Suggestions for Authors
the results need more work, it is not clear mainly Figure 2
Author Response
- The results need more work, it is not clear mainly Figure 2
Author Response: Thank you for your feedback. We have revised the results section to provide a clearer and more detailed explanation of Figure 2. We have added specific descriptions of the trends related to obesity, modifiable risk factors (such as smoking, physical inactivity, and diet), and intermediate risk factors, as well as major chronic diseases over the four-year period.
"Table 2 illustrates the distribution of participants across both groups (with and without obesity) for eight chronic diseases. Participants with obesity exhibited higher incidence rates for all types of diseases, including hypertension, hypercholesterolemia, type 2 diabetes, cardiovascular diseases, lung diseases, cancer, stroke, and genetic disorders. Over the four-year period, the percentage of individuals diagnosed with hypertension and obesity decreased from 25.4% in 2020 to 20.2% in 2023, with a corresponding reduction in individuals without obesity from 12.8% to 8.8%. A similar trend was observed for hypercholesterolemia, with obesity-related diagnoses decreasing from 25.0% in 2020 to 21.6% in 2023. In contrast, diagnoses of hypercholesterolemia in individuals without obesity remained relatively stable, ranging from 9.5% to 10.7%. In the case of type 2 diabetes, the proportion of individuals with obesity diagnosed with the condition decreased from 22.5% in 2020 to 18.8% in 2023, while the percentage among those without obesity remained stable around 10%. Cardiovascular diseases followed a similar trend, with the proportion of individuals with obesity diagnosed decreasing from 8.4% in 2020 to 7.3% in 2023. In individuals without obesity, the prevalence of cardiovascular diseases remained steady at around 4%. Respiratory diseases showed a slight decline among individuals with obesity, decreasing from 11.4% in 2020 to 9.4% in 2023, while the proportion of individuals without obesity remained around 7%. Cancer diagnoses among individuals with obesity also showed a small decrease from 2.6% in 2020 to 2.0% in 2023. Stroke diagnoses remained relatively low but stable, with the percentage of individuals with obesity diagnosed with stroke slightly decreasing from 3.0% in 2020 to 2.1% in 2023.
Lastly, genetic diseases were consistently more prevalent among individuals with obesity compared to those without, though the percentage decreased from 9.6% in 2020 to 6.6% in 2023. Individuals without obesity showed a steady prevalence of around 4-6% during the same period.
Table 3 presents the distribution of behavioral characteristics by obesity status, detailing patterns in smoking (cigarettes, waterpipes, and vaping), engagement in physical activities (both intensive and moderate), consumption of healthy foods (vegetables and fruits), and mental health risks (depression and anxiety). Smoking habits showed that the majority of participants, both with and without obesity, reported never smoking cigarettes, with a slight increase in the proportion of non-smokers among those with obesity, rising from 84.0% in 2020 to 86.7% in 2023. Similarly, waterpipe smoking (shisha) and vaping remained relatively low, with the majority of participants reporting never using these substances. Among those with obesity, 85.5% had never smoked shisha in 2020, and this percentage increased slightly to 87.7% in 2023. Vaping behavior followed a similar trend, with 92.7% of participants with obesity in 2020 reporting never vaping, compared to 89.7% in 2023.
Weight measurement habits indicate that a significant proportion of participants, both with and without obesity, reported having weighed themselves recently. In 2021, 25.4% of participants without obesity and 22.9% with obesity had measured their weight in the past week, with these percentages remaining stable in subsequent years. However, more than 30% of participants with obesity had not weighed themselves in over a month by 2023.
Dietary habits, as measured by acceptable levels of vegetable and fruit consumption, showed slight variations over the years. Among participants with obesity, 84.0% reported not consuming acceptable levels of vegetables in 2020, compared to 85.5% in 2023. Fruit consumption followed a similar pattern, with 88.8% of participants with obesity reporting inadequate fruit intake in 2020, which slightly improved to 92.2% in 2023. Combining both measures of vegetable and fruit consumption, a large proportion of participants, regardless of obesity status, consistently reported not meeting recommended levels throughout the study period.
Physical activity levels showed that a high percentage of participants with obesity were below recommended levels of intensive and moderate physical activity. In 2020, 85.5% of participants with obesity were below the recommended level of intensive physical activity, increasing to 91.4% by 2023. A similar trend was observed in moderate physical activity, with 83.2% of participants with obesity falling below recommended levels in 2020, and 87.8% in 2023. When combining both measures of physical activity, 67.5% of participants with obesity in 2020 were below recommendations, rising to 84.0% in 2023.
Mental health indicators, including risk of depression (measured by PHQ-2) and anxiety (measured by GAD-2), were added to the survey in 2021 and 2022, respectively. In 2021, 20.3% of participants with obesity were at high risk of depression, a figure that remained consistent at 20.7% in 2023. The proportion of participants with obesity at high risk of anxiety was 21.7% in 2022, slightly decreasing to 20.1% in 2023.
Figure 2 illustrates the visualization of trends related to obesity, modifiable risk factors, intermediate risk factors, and major chronic diseases from 2020 to 2023."
Reviewer 3 Report
Comments and Suggestions for Authors
This paper is a secondary analysis of data from a national surveillance survey. Four years of data were included in the data analysis. The paper is well-written with minor errors. My main suggestion is regarding the results. The study employed descriptive statistics only with raw numbers and percentages. Have you considered: 1) adding 95% CI, 2) adding inferential statistics to evaluate associations between obesity and independent variables?
Minor errors:
Line 23 chronic diseases?
line 95 Psychological or physiological?
Line 99 This study used secondary data obtained from.. or This study conducted a secondary data analysis using data obtained from...
Line 240 Based on the text after the sentence, self-weighing should be a protective factor, not a contributing factor. A contributing factor suggests negative consequences and a protective factor suggests beneficial effects.
Line 262, "however," suggests a contrast or opposition. Don't think it is appropriate in the context. Suggest deleting it.
Line 267. add a period.
Author Response
- This paper is a secondary analysis of data from a national surveillance survey. Four years of data were included in the data analysis. The paper is well-written with minor errors. My main suggestion is regarding the results. The study employed descriptive statistics only with raw numbers and percentages. Have you considered: 1) adding 95% CI, 2) adding inferential statistics to evaluate associations between obesity and independent variables?
Author Response: Thank you for your valuable comments. This paper aimed to explore the trend of obesity over a 4-year period. Previously, we published a paper on obesity and its associated factors in Saudi Arabia. In that study, we conducted association analyses and found similar results with this study. Therefore, we decided not to include the association analyses in this paper to avoid duplication, as it would not add new insights to the existing literature. The focus of this paper is on describing the trends over time rather than replicating previously reported associations. The reference paper can be found at https://www.mdpi.com/2227-9032/9/3/311
Minor errors:
- Line 23 chronic diseases?
Author Response: Noted with thanks. We have updated in the manuscript.
- line 95 Psychological or physiological?
Author Response: Noted with thanks. We have updated the manuscript and corrected it to the appropriate term.
- Line 99 This study used secondary data obtained from.. or This study conducted a secondary data analysis using data obtained from...
Author Response: Noted with thanks. We have revised the sentence to read: " This study conducted a secondary data analysis using data obtained from the Sharik Health Indicators Surveillance System (SHISS). The SHISS is a biannual, national cross-sectional survey implemented across Saudi Arabia through telephonic interviews".
- Line 240 Based on the text after the sentence, self-weighing should be a protective factor, not a contributing factor. A contributing factor suggests negative consequences and a protective factor suggests beneficial effects.
Author Response: Thank you for your observation. We agree with your assessment and have revised the sentence accordingly. Self-weighing is now referred to as a protective factor rather than a contributing factor, to better reflect its beneficial effects as discussed in the text.
" This trend is notable as self-weighing has been implicated as a protective factor to obesity in several preliminary studies [52, 53]."
- Line 262, "however," suggests a contrast or opposition. Don't think it is appropriate in the context. Suggest deleting it.
Author Response: Thank you for your suggestion. We agree that "however" may not be appropriate in this context, and we have deleted it from the sentence as recommended.
- Line 267. add a period.
Author Response: Noted with thanks. We have added the missing period at the end of the sentence in line 267 as suggested.
Round 2
Reviewer 1 Report
Comments and Suggestions for Authors
Thank you for improving the manuscript.